# Gait Analysis Using Accelerometry Data from a Single Smartphone: Agreement and Consistency between a Smartphone Application and Gold-Standard Gait Analysis System

**DOI:** 10.3390/s21227497

**Published:** 2021-11-11

**Authors:** Roy T. Shahar, Maayan Agmon

**Affiliations:** The Cheryl Spencer Institute of Nursing Research, University of Haifa, Haifa 3498838, Israel; rtzemahs@campus.haifa.ac.il

**Keywords:** gait analysis, inertial measurement unit, wearable sensors, smartphone, validation

## Abstract

Spatio-temporal parameters of human gait, currently measured using different methods, provide valuable information on health. Inertial Measurement Units (IMUs) are one such method of gait analysis, with smartphone IMUs serving as a good substitute for current gold-standard techniques. Here we investigate the concurrent validity of a smartphone placed in a front-facing pocket to perform gait analysis. Sixty community-dwelling healthy adults equipped with a smartphone and an application for gait analysis completed a 2-min walk on a marked path. Concurrent validity was assessed against an APDM mobility lab (APDM Inc.; Portland, OR, USA). Bland–Altman plots and intraclass correlation coefficients (agreement and consistency) for gait speed, cadence, and step length indicate good to excellent agreement (ICC_2,1_ > 0.8). For right leg stance and swing % of gait cycle and double support % of gait cycle, results were moderate (0.52 < ICC_2,1_ < 0.62). For left leg stance and swing % of gait cycle left results show poor agreement (ICC_2,1_ < 0.5). Consistency of results was good to excellent for all tested parameters (ICC_3,1_ > 0.8). Thus we have a valid and reliable instrument for measuring healthy adults’ spatio-temporal gait parameters in a controlled walking environment.

## 1. Introduction

Gait is a fundamental daily function crucial for independence in older age [1]; as such, it is a widely investigated marker in relation to aging [1], cognition [2], walking abilities, and health in general [3]. Spatio-temporal gait parameters can provide both valuable health information and recognition of divergence from normal patterns. It may also help identify an underlying pathology or measure its progression [4]. Modern gait analysis methodology has evolved from the days in which it meant a restricted clinic or research space using a pressure mat [5] or optical motion-capture systems [6]. Insoles [7] or other wearable devices now allow long-term assessment of gait; these systems rely on wireless body fixed sensors that include inertial measurement units (IMU) recording acceleration-derived measures of gait [4].

To date, gold-standard gait analysis systems include optical motion-capture systems, force plates, or expensive instrumented walkways. They also require high proficiency and both a long set-up time and manual post-processing that limits their accessibility and feasibility in both clinical practice and research [8]. By contrast, wearable systems are promising [7,8,9]. Remote, home-based evaluation can provide healthcare professionals with valuable and accessible information regarding patients’ clinical progress [10] and may pave the way for gait analysis in larger population cohorts for whom the current gold-standard methods are impractical. Wearable IMU is cheaper and more accessible [7].

Use of smartphones, in addition, has become common: pedometers counting the number of daily steps can be replaced with a smartphone application [11]. In addition, integrated hardware accelerometers do more than measure the number of steps. Smartphone kinematic hardware can address other health measures, such as postural control [12] and even gait [13]. Smartphones provide a portable, wearable, and cheap method that can be applied anywhere, not limited to a clinic environment or a clinician’s supervision and operation. 

A study published in 2017 demonstrated that a smartphone attached to the ankle and an algorithm the study team developed were feasible methods to perform gait analysis (calculating stride time, stance time, swing time, and cadence) [14]. Use of a smartphone placed in an individual’s pocket was also reported, showing that smartphone-measured gait analysis (stride time only) can be achieved with no abnormal positioning of the device [15]. These reports align with previous attempts made with old generation smartphones [16,17]; however, all provide limited usability in wide population studies or require special positioning of the smartphone device to obtain gait spatio-temporal data. Both a systematic review focusing on use of wearable sensors published in 2020 [18] and a search in current literature for publications focusing on commercially available smart-device use for gait analysis revealed that regardless of its convenience, its involvement in large population studies or remote home-base evaluation is still rare. 

The use of a widely available and affordable smartphone device with the newest IMU hardware is promising, possibly offering accurate analysis of gait equivalent to data obtained using a gold-standard gait analysis system. Developing algorithms using a smartphone carried naturally in a pocket to analyze gait in a wide population paves the way to a new era in gait analysis methods. The aim of this study was to evaluate a new algorithm (OneStep application) providing an elaborated gait analysis based on kinematic data collected by a single smartphone positioned in a user’s pocket. Concurrent validity was measured with a gold-standard wearable sensors-based gait analysis system (APDM mobility lab [19]) for measuring spatio-temporal gait parameters among community dwelling adults.

## 2. Materials and Methods

### 2.1. Setting and Study Population

The study was approved by the ethics committee of the faculty of social welfare and health sciences in the University of Haifa, Israel; participants were recruited using social media advertising and snow-ball recruiting. Inclusion criteria (1) age > 18; (2) ability to walk unassisted for up to two continuous min and (3) provide informed consent to participate. Exclusion criteria included (1) acute illness; (2) musculo-skeletal disorder or neurologic conditions affecting mobility; and (3) pregnancy. 

All procedures described herein were performed in an open public space where a participant’s privacy was kept (side corridor in a university building) or in the participant’s personal home. A licensed physical therapist (RTS) conducted all study procedures; although all participants were community dwelling independent adults, with no orthopedic or neurologic restrictions, in order to keep participant’s safety, the study procedure was closely supervised by a physical therapist. After oral explanation of study purpose and requirements, each participant signed an informed consent followed by recording of his/her age, gender, and height.

### 2.2. Gait Analysis Methods

Gait performance was measured using APDM Mobility Lab; a sensor-based system (APDM Inc., Portland, OR, USA) as a reference standard [19]. The system includes three IMUs attached with straps on both feet and the fifth lumbar vertebra (Figure 1). Each of these IMUs includes two tri-axial accelerometers, a gyroscope, and a magnetometer and records at a sampling frequency of 128 Hz. Additional information on the system can be obtained elsewhere [20]. The system wirelessly transmits collected kinematic data to a personal computer by radio-frequency communication through an access point, and collected data analysis is performed by dedicated software. Spatio-temporal gait parameters derived from the automatic output of the manufacturers include cadence (steps/min), speed (m/s), stride length (m), % double support phase from gait cycle, % swing phase from gait cycle, % stance phase from gait cycle and more. 

An application using a smartphone embodied inertial measurement units in order to obtain gait analysis spatio-temporal gait information (OneStep software application, version 2.9, Celloscope Ltd., Tel Aviv, Israel). We used a single smartphone (Android device) deriving all kinematic data from the motion of a single leg; to ensure similar positioning of the smartphone by all participants, we used a strap and a plastic pocket positioned on the right anterior thigh (Figure 1). The smartphone inertial measurement unit provides a sequence of 3D acceleration, angular velocity, and magnetic intensity data at a sampling rate of 100 Hz. To note, similar commercially available IMU for gait analysis (DynaPort MT; McRoberts B.V., The Hague, The Netherlands) offer similar hardware for 3D accelerometery and angular velocity accompanied by a magnometer, with a sampling frequency of 100 Hz, comparable to the smartphone hardware. Indeed, while choosing a specific device for measurement, the clinician or researcher should consider several parameters, such as costs, ease of use, and the spectrum and accuracy of parameters obtained. In this regard, the OneStep application has the potential to offer an accessible and effective solution for clinicians and researchers.

### 2.3. Study Procedure

A walking path 8 to 10 m long was measured using a tape and marked with plastic cones. After placing body-worn sensors and a smartphone running the OneStep application in the plastic pocket, each participant was asked to walk in the marked path for two min, then to walk in one of four walking patterns (normal pace, fast pace, slow pace, and walking while performing a cognitive task). A specific walking pattern was randomly assigned; this method was employed to extend variability of gait patterns and challenge the ability of the OneStep application to measure different walking styles to potentially represent various types of walking in the general population. Specific instructions assured altered gait patterns: “walk at a comfortable speed” for normal pace walking, “walk as fast as you can” or “walk as if the floor is slippery” for fast and slow pace respectively, and while continuously subtracting the number seven from a given number, for dual task walk. The systems (i.e., APDM mobility lab and OneStep application) recorded the walk simultaneously; to ensure adequate analysis of spatio-temporal data, walks where APDM mobility lab reported collecting less than 15 gait cycles were repeated. 

### 2.4. Data Collection

Spatio-temporal gait parameters derived from the automatic output of APDM mobility lab software included gait speed, cadence, stride length, and gait double support phase % (for each, the manufacturer’s output included two measures, one from the left foot and one from the right foot, that we averaged to a single measure to allow comparison to the OneStep application output), and gait stance phase % and swing phase % for each leg (right and left), for a total of eight outcome variables. The kinematic data collected with the OneStep application was uploaded to a cloud server for analysis; results were prepared and sent to the study team by the company developing the application in an electronic spreadsheet. To note—data collected by APDM mobility lab was kept separately on a local computer and was blinded to anyone but the study team; any personal information or other personal identification of participants was kept on consent forms only and was never uploaded to OneStep servers.

### 2.5. Statistical Analysis

Statistical analysis was performed using Jamovi (a free and open statistical platform for statistical computing, version 1.6.23.0) [21] with two added modules for intraclass correlation calculation [22] and Bland–Altman plots [23] to visually display agreement and consistency between system pairs. 

Intraclass Correlation Coefficients (ICC) were calculated as standardized measures of agreement, using a two-way random effect model for absolute agreement (ICC_2,1_), and a two-way mixed-effect model for consistency (ICC_3,1_) [24]. Following suggested rule of thumb [24] and recommendations for sample size when measuring ICC [25], sample size was set at 60 heterogeneous samples, with statistical significance for an alpha-value as 0.05 and a power of greater than 80%. Under such conditions, result ICC < 0.5 is indicative of poor reliability, 0.5 ≤ ICC < 0.75 is considered moderately reliable, 0.75 ≤ ICC < 0.9 indicates good reliability and values greater than 0.90 indicate excellent reliability [24].

## 3. Results

### 3.1. Participant’s Characteristics

The study sample included 60 adults, 18 to 80 years (mean 37.2 ± 13.4 years); 52% were women. Participants’ height ranged from 150 to 191 cm (mean 171 ± 10 cm), for a total of 60 paired walks. A total of 60 walks were recorded with 15 participants performing four walking patterns. To note, seven walks needed to be repeated due to low number of analyzed gait cycles reported by APDM mobility lab software (all of which were fast or slow-paced walking).

### 3.2. Concurrent Validity

Good to excellent absolute agreement between APDM mobility lab and OneStep application was found for gait speed, cadence, and stride length (ICC_2,1_ 0.80–0.99) (Table 1). Stance right%, swing right% and double support% showed moderate absolute agreement (ICC_2,1_ 0.52–0.62) and lastly, left stance% and left swing% showed poor absolute agreement (ICC_2,1_ 0.4 for both). Despite observed variability in absolute agreement, all measures showed good to excellent consistency expressed through larger ICC_3,1_ (0.83–0.99). We found the highest levels of agreement and consistency for gait speed, cadence, and stride length. Bland–Altman plots for the agreement between APDM mobility lab and OneStep application are provided in Figure 2a–h.

## 4. Discussion

The aim of this study was to evaluate the concurrent validity of a single smartphone hardware IMU placed in a pocket positioned on the right leg for gait analysis. For evaluating concurrent validity, a gold-standard high-cost multi-sensor APDM mobility lab was used as external reference. Good to excellent absolute agreement occurred between them with the latter being a significantly lower-cost, simple to use IMU system for measuring gait speed, cadence, and stride length (ICC = 0.8–0.99). ICC consistency values were good to excellent (0.83–0.99) in all recorded measures (gait speed, cadence, stride length, stance % of gait cycle for both right and left legs, swing % of gait cycle for both right and left legs and double support % of gait cycle). 

Previous attempts to evaluate the potential use of a single commercially available IMU designed for gait analysis to measure gait spatio-temporal parameters reported similar results: good to excellent concurrent validity of measured gait parameters [8,26,27]. However, all positioned the IMU on the lumbar spine to track the kinematics of the center of body mass. Similarly, the use of an older-generation commercial smart-device (iPod Touch) placed on the lumbar spine to measure acceleration and position for gait analysis also reported good to excellent ICC for position data and moderate ICC for acceleration data [28]. An additional study used a newer smartphone (iPhone 4s) placed on either the lateral side of the hip or the lumbar spine, comparing to a different motion-capturing gold-standard system [29], concluding good accuracy of smartphone estimation (report limited to step count, heel strike detection, step time and length). Although placing the IMU on the lower back or ankle seems simple enough, accurate positioning may require assistance or supervision; thus, the proposed method of placing the smartphone in a front-facing pocket provides a desired ease of use.

Although a smartphone hardware IMU (iPhone) placed in a front pocket was previously reported to provide accurate stride time data during a normal and dual-task walk, the results were limited to this single parameter [15]. The current study, to the best of our knowledge, is the first to evaluate a wide-array of spatio-temporal gait parameters collected by a single smartphone placed naturally in a front pocket, following the recommendations made in a systematic review on different gait analysis methods published in 2020 [18]. 

The discrepancy between results of absolute agreement (ICC_2,1_) and consistency (ICC_3,1_) of spatio-temporal gait parameters (stance % of gait cycle, swing % of gait cycle, and double support % of gait cycle) measured by the two gait analysis methodologies suggests that the two measurement systems agree on which walks have higher and lower values measured parameters. Yet there is a consistent bias between the absolute measurements of the two systems that causes one to always predict a value that is a set amount higher or lower than the prediction of the other system. Observation of the Bland–Altman plots confirms this—the error for each parameter is distributed symmetrically around the mean error and the limits of agreement are lower than the range of the measured parameters, but the mean error is significantly different than 0.

High consistency with lower absolute agreement, of a similar magnitude to that which we observed in this study, is common when evaluating gait analysis systems, even when comparing existing gold-standard systems [9]. Such differences could arise because the method by which each system detects gait event leads to different interpretations of the time of such events [27]. For example, an intertial-based system like APDM mobility lab might detect the terminal contact of a foot during a stride as the moment at which the foot starts moving up and forward, while a pressure-based system may instead detect terminal contact as soon as the foot stops applying enough pressure on the ground. Future validation studies of the OneStep software application gait analysis system comparing it to other gold-standard systems will further evaluate the accuracy of the system and determine which current gold-standard system it most resembles in its absolute measurements of spatio-temporal gait parameters.

To note, absolute agreement between spatio-temporal parameters of the right leg (where the smartphone was placed) was higher than that of the opposite side (ICC_2,1_ of right swing % of gait cycle was 0.621, relatively higher than that of the left side, ICC_2,1_ = 0.404) suggesting that performing the analysis with two smartphones simultaneously may provide even better results.

This study highlights the simplicity of a smartphone application for gait analysis, possibly replacing the high costs, space demands, and time-consuming efforts required by current gold-standard systems with a simple and cheaper method. Smartphone IMUs do not require calibration of wearable sensors prior to use and remove the barrier of needed training for different software operations—simply place a smartphone with the application in a pocket and walk for a few minutes and clinically valuable spatio-temporal gait analysis data can be obtained. Furthermore, for larger sample sizes or population-based studies where it is impractical to use most gait analysis systems, use of a smartphone application makes purchasing any other commercially available IMU’s redundant; smartphone-based gait analysis may provide an alternative that can be used anywhere (home or community) and anytime (for short or long periods of time) thus providing valuable clinical data on prospect study population and even distant patients: assessing individual’s performance remotely, to monitor changes and support remote treatments [30].

## 5. Limitations

The current study followed recommended instructions provided by an APDM mobility lab. Thus, the results of the current study are limited to a controlled lab environment (participants walked in a straight line on an indoor surface). OneStep application use is not limited to these conditions, suggesting that different settings may provide different results. Additional validation studies, comparing the OneStep application to other gold standard IMUs, is needed to establish its validity further.

Additional validation of the OneStep application using different gait analysis systems, such as optical motion-capture systems or pressure mats, is still required to establish application-based gait analysis method as equivalent to other gold-standard gait analysis systems.

Although sample size was adequate, it was relatively small; we included variability of walking types and mixed participant demographics (age, sex, and height). All participants were healthy. However, we did not compare the psychometric properties of different walking styles or other participant’s characteristics (e.g., body mass index, height, or sex). Additional validation study is needed to assess the OneStep application for gait analysis in different conditions and populations (e.g., outdoor walking or longer walking period and different subgroups with gait pathologies such as Parkinson’s disease or cerebrovascular accidents), while comparing the psychometric properties of the measurements between these populations.

Comparing the OneStep application to different, single IMU (such as the validated DynaPort MT mentioned above) can provide valuable information on gait analysis performed in natural settings, such as home environment or outdoors. As this method mostly resembles the use of smartphone placed in a pocket it may provide the substantial data needed to assure that a smartphone can replace the commonly used community-based device for gait analysis.

## 6. Conclusions

The results of this study showed that the OneStep application software as a single IMU system for gait analysis has good to excellent concurrent validity (consistency) of all measured gait parameters compared with the three wearable sensors APDM mobility lab. Results showed that for gait speed, cadence, and stride length absolute agreement between OneStep and APDM mobility lab was good to excellent, for stance right % of gait cycle, swing right % of gait cycle and double support % gait cycle showed moderate absolute agreement and for left stance and swing % of gait cycle absolute agreement was poor.

Overall, the OneStep application for gait analysis provides a highly portable, feasible, and easy-to-use method to assess patient status that can provide clinicians and researchers with a set of valid, reliable, and sensitive spatio-temporal gait parameters for quantitative gait analysis.

## Figures and Tables

**Figure 1 sensors-21-07497-f001:**
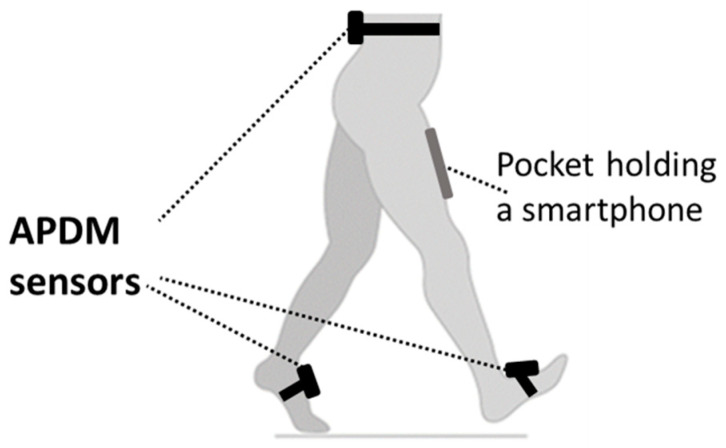
IMU setting. APDM mobility lab sensors were placed over participant’s clothing and were wrapped around the shoes to secure the sensor to the participant’s body. The plastic pocket used to hold the smartphone was fixed using belt-like Velcro straps.

**Figure 2 sensors-21-07497-f002:**
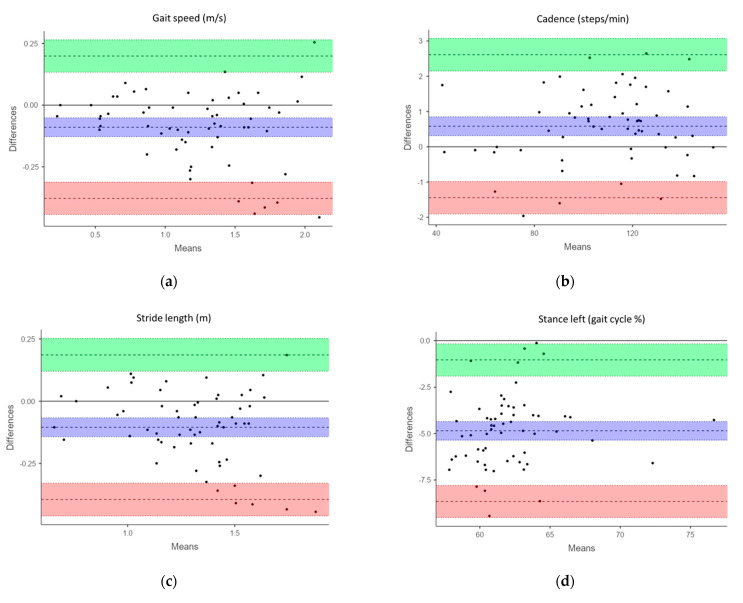
(**a**–**h**): Bland-Altman plots (taken directly from the Jamovi platform) for (**a**) gait speed, (**b**) cadence, (**c**) step length, (**d**) stance left gait cycle %, (**e**) stance right gait cycle %, (**f**) swing left gait cycle %, (**g**) swing right gait cycle %, and (**h**) double support gait cycle %. Dashed lines indicate upper and lower 95% limits of agreement (±1.96 SD of the bias).

**Table 1 sensors-21-07497-t001:** Summary of results.

	APDM Mean ± SD	OS Mean ± SD	ICC Consistency	ICC Agreement (95% CI)
Speed (m/s)	1.18 ± 0.453	1.27 ± 0.487	0.951	0.935 (0.896–0.962)
Cadence (steps/min)	109 ± 25.8	108 ± 25.7	0.999	0.999 (0.998–0.999)
Stride length (m)	1.26 ± 0.255	1.36 ± 0.296	0.857	0.801 (0.688–0.871)
Stance left (gait cycle %)	59.7 ± 3.45	64.5 ± 3.22	0.830	0.404 (0.182–0.570)
Stance right (gait cycle %)	59.4 ± 3.51	62.8 ± 3.48	0.900	0.621 (0.381–0.748)
Swing left (gait cycle %)	40.3 ± 3.45	35.5 ± 3.22	0.830	0.404 (0.171–0.580)
Swing right (gait cycle %)	40.6 ± 3.51	37.2 ± 3.48	0.900	0.621 (0.369–0.741)
Double support (gait cycle %)	19.2 ± 6.77	27.3 ± 6.39	0.918	0.524 (0.290–0.680)

## Data Availability

Raw data file is publicly available https://doi.org/10.5281/zenodo.5656346 (accessed on 13 October 2021).

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
