# Peer review of "Gait Analysis Using Accelerometry Data from a Single Smartphone: Agreement and Consistency between a Smartphone Application and Gold-Standard Gait Analysis System"

_sensors, 2021, doi:10.3390/s21227497_

Round 1
Reviewer 1 Report
The authors have presented their research about Gait analysis using accelerometry data from a single smartphone: agreement and consistency between a smartphone application and gold-standard gait analysis system.
The authors have put forward a fine effort in presenting their research.
In the view of maintaining the scope and quality of the Journal, there are some minor concerns that must be addressed before this article can be accepted for publication.
Below are the comments and suggestions to improve the article:
- As indicated in the multiple references within the article and numerous other articles published already, there are numerous other approaches that attempts to achieve a similar objective.Please include a section to emphasize how the approach of this research is different from others, mainly through technical comparison (for example: technology comparison table).
- Since human subjects can be involved in such studies, the authors can include a section about their views on sample size requirements, ethical approval and/or consent requirements, general guidelines about safety, methods and study design, experimental controls, bias in algorithm models, etc., for performing human subject studies and using the available data. This could be of great interest to the readers and researchers as this is usually not covered in other articles despite the importance.
- Please make sure all the figures have references/citations (throughout the article)
- Please make sure references/sources are adequate and correctly used (throughout the article)
- Please review usage of abbreviations/acronyms, and provide an explanation where required (throughout the article)
- Some grammatical and sentence structure related revisions are required. The authors are requested to check for missing words, identifiers, and dangling modifiers throughout the manuscript to improve the consistency and quality.
- Please make sure the text in the figures is legible. For example, figure 2 needs improvement.
Author Response
Thank you for your valuable input and effort in reviewing the manuscript. Here are our responses to your feedback; keep in mind that line numbers are for viewing the manuscript file with “track-changes” on to allow for easier review of changes made.
- We further clarified the aim of this study, which was to compare a smartphone application to a validated gold-standard gait analysis method (lines 68-72). However, in order to be more inclusive, we added additional information comparing the smartphone hardware to a different inertial measurement unit (Dynaport MT). See lines 121-128.
- Concerning safety of human participants, a clarification was made in lines 94-96. additional justification for sample size was also presented, with a new citation focusing on sample size for studies measuring ICC (lines 171-173).
- Manuscript was sent to a professional editor (Dr. Jenn Lewin) to assure adequate format and use of abbreviations, citing and grammar. Corrections were made throughout the manuscript.
- See section 3.
- See section 3.
- See section 3.
- Figures 2a-2h quality was improved by providing a separate file for each image, for the final version of the manuscript.
Reviewer 2 Report
In this work, the authors investigated the concurrent validity of a smartphone, which is placed in a front facing pocket, to perform gait analysis. A lot of similar work was published. This work does not offer much innovation or a unique idea. It feels like a very junior job. There is no suggestion other than to reject the draft.
Author Response
We value your input, we wish to clarify that the focus of the current study is not to investigate the concurrent validity of a smart phone but the concurrent validity of a specific application that may be used for population-based studies and in clinics. We understand that we did not clarify this point enough, thus we have further refined it in lines 46-51. Additionally, the need or gap in the literature was further explained in lines 68-71.
Reviewer 3 Report
Please discuss how the altered gait patterns affected the results. Were those partitions visible on graphs?
Please discuss how to remove/minimize the biases reported in the study.
Suggest provide more information in the figure captions.
Author Response
Thank you for your valuable input and effort in reviewing the manuscript. Here are our responses to your feedback; keep in mind that line numbers are for viewing the manuscript file with “track-changes” on to allow for easier review of changes made.
The aim of using different types of walking was to present the general population and the ability of the application to capture them. We have further emphasized this point in lines 141-142. Due to relatively small sample size within each subgroup of walking patterns, power to detect differences is limited. We have added this limitation to the relevant section (lines 289-291).
We agree with your observation, an additional point was added to the limitation’s sections (lines 296-301) suggesting additional requirements needed to establish the use of smartphone application for community-based gait-analysis.
Additional information was added to both figures available in the manuscript.
Round 2
Reviewer 1 Report
You have rectified the errors and/or have addressed the suggestions and corrections cited in the previous review. This article can be accepted for publication pending any grammatical / language corrections and editorial procedures.
Reviewer 2 Report
Thanks to the authors for their responses. Then I wasn't convinced that the job was innovative enough. For scientific work, it's more about doing a small application.